# Relating Regularization and Generalization through the Intrinsic Dimension of Activations

**Bradley C.A. Brown**
University of Waterloo
bcabrown@uwaterloo.ca

**Jordan Juravsky**
University of Waterloo
jordanjuravsky@gmail.com

**Anthony L. Caterini**
Layer 6 AI
anthony@layer6.ai

**Gabriel Loaiza-Ganem**
Layer 6 AI
gabriel@layer6.ai

## Abstract

Given a pair of models with similar training set performance, it is natural to assume that the model that possesses simpler internal representations would exhibit better generalization. In this work, we provide empirical evidence for this intuition through an analysis of the intrinsic dimension (ID) of model activations, which can be thought of as the minimal number of factors of variation in the model's representation of the data. First, we show that common regularization techniques uniformly decrease the last-layer ID (LLID) of validation set activations for image classification models and show how this strongly affects generalization performance. We also investigate how excessive regularization decreases a model's ability to extract features from data in earlier layers, leading to a negative effect on validation accuracy even while LLID continues to decrease and training accuracy remains near-perfect. Finally, we examine the LLID over the course of training of models that exhibit grokking. We observe that well after training accuracy saturates, when models "grok" and validation accuracy suddenly improves from random to perfect, there is a co-occurent sudden drop in LLID, thus providing more insight into the dynamics of sudden generalization.

## 1 Introduction

Occam's razor states that given a set of plausible explanations, the simplest one should be preferred. The benefits of applying this idea to machine learning models are intuitive: any method whose decision criteria are too complex can learn to overfit to noise and biases in the data that are unrelated to the problem at hand. In contrast, methods that are able to fit the observed data with the fewest number of assumptions are more likely to have found generalizable patterns in the data applicable to unseen examples. The successful application of this idea can be seen throughout machine learning (e.g. [32, 1, 5, 28, 10, 9]). Perhaps most notably, Bayesian inference itself can be seen through the lens of Occam's razor [25], with the likelihood and prior encouraging plausibility and simplicity, respectively.

Our work can be seen as another application of Occam's razor, where we analyze how the simplicity and plausibility of learned neural network representations are affected by regularization and correlated with generalization. Specifically, we elucidate this idea using the intrinsic dimension (ID) of the validation data's activations, which can be thought of as the minimal number of factors of variation in their representation. We introduce two measures of representational compexity for neural networks: last-layer ID (LLID) and peak ID (PID). LLID is the intrinsic dimension of the validation set's last layer of activations and PID is the maximum intrinsic dimension of the validation set's activations

Has it Trained Yet? Workshop at the Conference on Neural Information Processing Systems (NeurIPS 2022).

over all layers of the model (which for all experiments is always at early layers of the network). As we will show, under the framework of Occam's razor, a neural network is considered "simpler" if it has a low LLID and is "plausible" if training accuracy and PID are sufficiently high.

Given the link between representational simplicity and generalization, the success of vastly over-parameterized neural networks [35] seems counterintuitive. We highlight two explanations for this apparent contradiction. Firstly, Ansuini et al. [2] and Recanatesi et al. [28] show that despite having a large ambient space, activations live on orders of magnitude lower dimensional manifolds. Secondly, although DNNs are trained to maximize performance on the training set, they are also regularized. Regularization techniques bias the model towards simpler and more general solutions, thus reducing overfitting. In this work we unify these two explanations, showing that common regularization techniques decrease the dimensionality of activation manifolds deeper in the network and allow image classification models to better generalize to the validation set. Nevertheless, one cannot arbitrarily decrease LLID through regularization and expect a guaranteed improvement in generalization. We hypothesize that this is partly because regularization also decreases the ID of initial layers in the network, and thus decreases PID, inhibiting sufficiently complex/varied features from being learned. We show that achieving good generalization requires a trade-off between decreasing the LLID and keeping the PID sufficiently high.

To strengthen our understanding of the relationship between simplicity of representation and generalization, we also study a task where models are able to learn to perfectly generalize. Specifically, we analyze the internal representations of small transformer models trained to perform modular division. Power et al. [27] found surprising behaviour on these tasks where well after training accuracy saturates at $100\%$, models are able to "grok" the problem and suddenly increase their validation performance from random guessing to $100\%$. Remarkably, we find a co-occurrent decrease in LLID when the model groks. This suggests that there is a strong relationship between finding generalizable patterns in the data and the model's simplicity of representation, further confirming the relevance of measuring LLID.

We hope that our findings linking widespread regularization techniques to measurable proxies for simplicity motivate the community to continue to study the implications of LLID and PID, both as indicators of when a model can be relied upon to generalize and as targets for optimization. Specifically, disentangling the effect of regularization techniques invites future work studying the theory of how LLID, PID, regularization and generalization are related to further improve regularization and optimization methods. We will release the code publicly upon acceptance to facilitate this future research.

## 2  Background and Related Work

**Intrinsic dimension**   The manifold hypothesis states that despite having high ambient dimension, data of interest lives on a manifold of much lower dimension [3]. This hypothesis has been experimentally verified for both image data and activations in deep neural networks [26, 2, 8]. The dimension of this manifold is referred to as the *intrinsic dimension* of the data, e.g. if the data lies on the unit circle in $\mathbb{R}^2$, then its intrinsic and ambient dimensions would be 1 and 2, respectively. For this study, we use an estimate of the intrinsic dimension of the last layer of activations as a proxy for the simplicity of representation of a neural network. As it is intractable to calculate the exact intrinsic dimension, we use the TwoNN estimator of Facco et al. [12], following the implementation described in Ansuini et al. [2]. This estimator assumes that the density is locally constant around all datapoints, and then leverages the observation that the ratio

$$\frac{\log\left(1 - F(\mu)\right)}{\log\left(\mu\right)} \tag{1}$$

approximates the intrinsic dimension, where $F(\mu)$ is the cumulative distribution of $\mu = r^{(2)}/r^{(1)}$: the ratio of the distance of each datapoint's second and first nearest neighbours. While not immediately intuitive, this result is obtained by explicitly deriving $F(\mu)$ in terms of $\mu$ and the ID. Using the known functional form of $F(\mu)$, it can then be easily verified that the ratio in (1) matches the ID [12]. In practice, the empirical cumulative distribution $F^{emp}$ is calculated from the data and the slope of a straight line through the origin, fit to the observed coordinates $\{(\log(\mu_i), -\log(1 - F^{emp}(\mu_i)))\}_i$, is used as the intrinsic dimension estimate. We use this procedure to obtain estimates $\hat{d}_\ell$ of the intrinsic

dimension of all the activation layers $\ell = 1, \dots, L$ in a neural network (using validation data unless otherwise specified), and define its LLID and PID as

$$\text{LLID} = \hat{d}_L, \qquad \text{PID} = \max_{\ell} \hat{d}_\ell. \qquad (2)$$

Although more recent ID estimators exist [14, 7, 23, 31, 22], we opt for this one as it is practically reliable [2] and computationally efficient for calculating ID over high ambient dimension activations.

**Intrinsic dimension of activations**     Although other measures of simplicity have been proposed [11, 13, 6], we choose intrinsic dimension as our proxy for representational complexity as previous works have found that activations live on non-linear manifolds and intrinsic dimension estimators have been insightful at revealing important properties of the geometric structure of data representations [2, 28, 21, 24, 16]. In particular: Ansuini et al. [2] show that LLID is decreased for larger networks and is correlated with generalization, Lee and Chung [21] use the ID of activations as an early stopping criteria for few shot embedding adaptation; and Ma et al. [24] use activation ID as a criterion for decreasing confidence in noisy labels to avoid overfitting.

Notably, Ansuini et al. [2] show that the estimated ID of activation manifolds shows a hunchback shape in classification models where the intrinsic dimension quickly peaks in early layers of the network and then continues to decrease until the final activation layer. We note that intrinsic dimension cannot increase across neural network layers, which are functions of only their previous activations, and we attribute this inconsistency to errors in the ID estimator. Since we only require that trends and ID comparisons are consistent (not exact), we find that this is nevertheless a useful estimator. In other words, even though applying a function cannot change the true ID, it can clearly change nearest neighbour distances and thus the ID estimate, which nonetheless remains a valid measure of the complexity of learned representations. We further the work of Ansuini et al. [2] (who also found this ID estimate appropriate) and provide support for the usefulness of measuring the ID of activations by showing how PID and LLID are affected by regularization, correlated with generalization and strongly coupled with the perfect generalization phenomena of grokking.

**Grokking**     Recent work has identified training regimes where models experience a period of rapid generalization long after the model has perfectly memorized the training dataset, also known as grokking. Power et al. [27] discovered this phenomenon by training two-layer transformer models on small procedurally generated algorithmic datasets. Notably, the amount of weight decay applied during training had a large impact on the number of steps required for the model to generalize; stronger regularization caused grokking to occur significantly earlier in training.

**Regularization**     Regularization has been critical to the success of deep learning methods [20, 30, 19]. Of particular interest to our work are regularization techniques that attempt to decrease the complexity of activations [4, 36]. Our work quantifies this decrease in complexity, demonstrates that prominent regularization techniques [30, 19] also have this effect, and calls for more targeted regularization techniques that are layer-dependent.

## 3   Experiments

We introduce introduce two measures of representational complexity: the (estimated) intrinsic dimension of the validation set's last layer of activations, and the maximum (estimated) intrinsic dimension of the validation set's activations over all layers in the network, both defined in (2). In this section, we analyze the effects of LLID and PID in image classification (Section 3.1) and transfomer (Section 3.2) models.

### 3.1   Regularization and Generalization in Image Classification

**Setup:**     To analyze the relationship between LLID and regularization we train ResNet-18 models [15] on the CIFAR-10 and CIFAR-100 [18] image classification datasets while independently sweeping over values of dropout and weight decay [30, 19]. Training details can be found in Appendix A. The LLID for these models is defined on the activations after the final average pooling layer, and the PID is consistently on the activations after the first ResNet block. We note that all models trained achieve perfect accuracy on the training set.

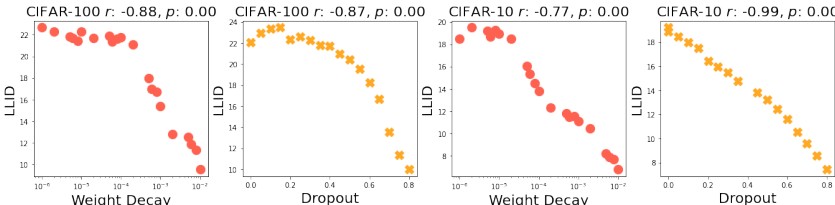

Figure 1: LLID for ResNet-18 models on CIFAR-100/CIFAR-10 datasets with varying values of weight decay and dropout percentage. Increasing regularization strength decreases LLID.

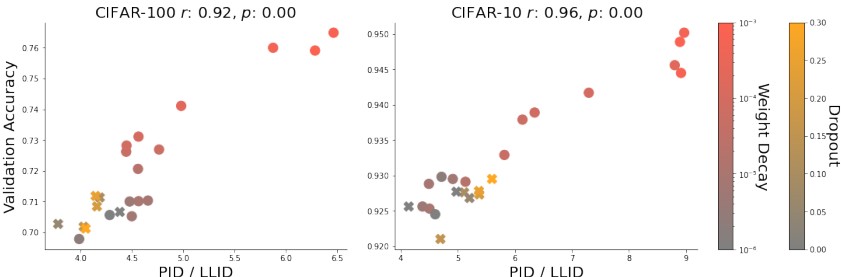

Figure 2: Relationship between the PID / LLID ratio and validation accuracy for models trained with various values of dropout (indicated with X markers) and weight decay (indicated with circle markers). Increased intensity of red and yellow color denotes increased weight decay value and dropout percentage respectively.

**Regularization decreases LLID:** In Figure 1, we plot the final LLID for models with different amounts of regularization at the end of training. Increasing the strength of the regularization, decreases LLID as confirmed by the Pearson's correlation coefficients, $r$, and $p$-values for independence in a $t$-test shown in the figure titles. This strong relationship confirms the intuition that regularization decreases the complexity of neural representations.

**Generalization is correlated with PID and LLID:** We have shown that decreasing LLID is one of the effects of regularization that is shared between both methods studied. However, different regularization techniques clearly have different consequences that may either aid or harm generalization and adjusting the strength of regularization requires a trade-off between these effects. We study one such negative impact that we observe in our models – decreasing PID – and leave a more detailed study of the other impacts of regularization to future work.

Figures 5 and 6 in Appendix B show that achieving higher validation accuracy is correlated with decreasing LLID and increasing PID. This indicates that although it is advantageous to have a simple representation of the data at deeper layers, it should not come at the expense of the ability to extract rich features in earlier layers. Figure 2 highlights this trade-off, showing the correlation between validation accuracy and the ratio between PID and LLID (i.e. how much the model reduces the ID of activations across layers). There is a clear trend where a higher PID/LLID ratio indicates better validation performance, highlighting the necessity of balancing the effect of decreasing the complexity of the last layer of activations (which are a single linear layer away from the logits and thus form the basis for the model's ultimate classification) without penalizing the model for extracting the potentially complex features in earlier layers required to form this representation.

The relevance of PID becomes more dramatic as the strength of regularization is pushed beyond normal bounds. In Figure 4 of Appendix B, we plot how PID changes with excessive regularization. We see that soon after regularization is increased beyond the value that gives optimal validation accuracy, there is a rapid decrease in both PID and validation accuracy. This is particularly true for the more challenging CIFAR-100 dataset where we hypothesize that the increase in the number of classes requires more varied features to be learned in earlier layers. Using the framework of Occam's Razor, we see that excessive regularization prevents "plausibility", and PID is a strong indicator of this occurrence.

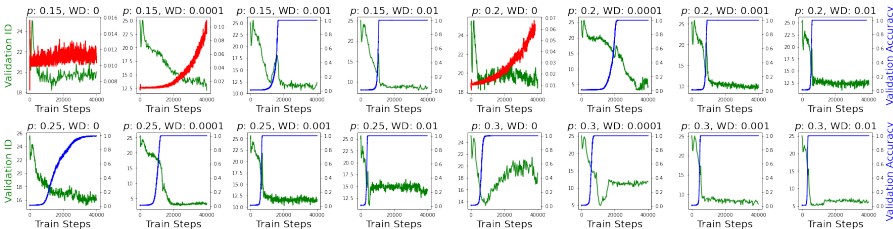

Figure 3: Validation accuracy (blue) and LLID (green) of transformer models throughout training on modular division tasks (plot titles show percentage of training data used, $p$, and weight decay coefficient, $WD$). We observe co-occurent ID drops when transformers 'grok'. Accuracy of runs where model does not achieve perfect validation accuracy in the allotted compute budget of 40k training steps is shown in red.

## 3.2 LLID and Grokking

**Setup:** We last explore the relationship between LLID and the grokking phenomenon observed in Power et al. [27]. Grokking characterizes a sudden rise in validation performance long after the model has perfectly memorized the training dataset. This effect was specifically observed on a small algorithmic dataset using a two-layer transformer model [33]. We reproduce this training setup while monitoring training and validation set LLID over the course of training. We train 16 different models, sweeping over the strength of weight decay regularization and the size of the training dataset, since these parameters strongly affect when models begin to grok and the duration of the grokking period. For these two-layer transformer models, we define the LLID as the intrinsic dimension of the activations after the second transformer block's MLP (we also measure the ID of activations after the first transformer block and find similar patterns which we hypothesize is due to the small network size). More training details can be found in Appendix A.

**LLID drops co-occur with grokking:** In Figure 3, we observe a sudden drop in LLID co-occurent with the sudden rise in validation accuracy observed in models that exhibit sudden generalization (we show the same plot alongside the training LLID and accuracy of each run in Figure 7 of Appendix B). Notably, models that grok have an average minimium LLID of $8.53$, much lower than the average minimum LLID – $16.16$ – of models that remain at a low validation accuracy in the allotted compute budget, once again confirming the relevance of the simplicity of representations for generalization as measured by LLID. Curiously, we also observe that for lower amounts of training data when grokking occurs later in training (ex. plot with $p = 0.15$, weight decay$= 0.01$ in Figure 3), there is an initial LLID spike when the model begins grokking followed by a descent into a lower LLID than the previous plateau. This hints at an escape from a local minima, and the onset of this phenomena at higher weight decay values indicates that regularization helps with this jump. This relationship between LLID and grokking is exciting, as it both confirms the validity of measuring simplicity using LLID and shows that it is directly correlated with generalization. Moreover, optimizing for LLID with the aim of replicating this increase in generalization is an exciting direction for future optimization and regularization methods.

## 4   Conclusion

In this work we explored the link between internal representation simplicity and generalization performance by analyzing the intrinsic dimension of model activations. We identified a strong correlation between increasing amounts of regularization with decreased LLID in image classification models. We also demonstrated that decreased LLID and increased PID correlates with improved generalization, until a point is reached where excessive regularization reduces the capability of models to extract meaningful features in earlier layers. Lastly, we established a co-occurence between a sudden rise in validation accuracy (i.e. grokking) and a sudden drop in LLID on small models trained on algorithmic datasets. Overall, our results show a strong correlation between the intrinsic dimension of activations and generalization. We hope this will lead to increased interest to understand this phenomenon from a theoretical perspective and untangle the role regularization has in intrinsic dimension through optimization.

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

# A  Training Details

**CIFAR Image Classification**   For our regularization sweeps, we independently sweep over weight decay with values of $ae^{-b}$ for all combinations of $a \in \{1, 2, 5, 6, 8\}$ and $b \in \{-6, -5, -4, -3, -2\}$ as well as dropout [30, 19] in between each residual block with dropout probabilities ranging from $0.05$ to $0.8$ with increments of $0.05$. At the end of training, we estimate the intrinsic dimension across all validation datapoints after each of the ResNet blocks (when the channel dimension changes) and final average pooling layer. To calculate the intrinsic dimension, we flatten the shape of the activations and calculate ID on this vector using the TwoNN ID estimator [12]. To train all models, we use the SGD optimizer with a momentum value of $0.9$ and a learning rate of $0.1$ decayed by a factor of $0.2$ at epochs $60, 120$ and $160$. We train for a total of $200$ epochs. The data is augmented by applying random crops, random horizontal flips with probability of $0.5$ and random rotations between $-15$ and $15$ degrees.

**Modular Division Tranformer Models**   To train all models, we use a 2 layer transformer decoder model with $4$ heads and an embedding dimension of $128$ obtained from the huggingface transformers library [34, 33]. The ADAM optimizer [17] is used with beta values of $(0.9, 0.98)$ and a constant learning rate of $0.001$ for $40k$ training steps. The modular division modulo 97 dataset is obtained from [29].

# B  Additional Figures

Due to space constraints, in this section we highlight additional results.

**CIFAR Results:**   In Figure 4 we show the effect of excessive regularization on the PID of ResNet-18 models. We note that as regularization is pushed beyond typical bounds, PID and validation accuracy decrease. In Figure 5 and 6 we further decompose the relationship shown in Figure 2 by displaying the correlation between LLID and PID with validation accuracy respectively.

**Grokking Results:**   In Figure 7 we show a larger and more detailed version of Figure 3 including both training accuracy and LLID.

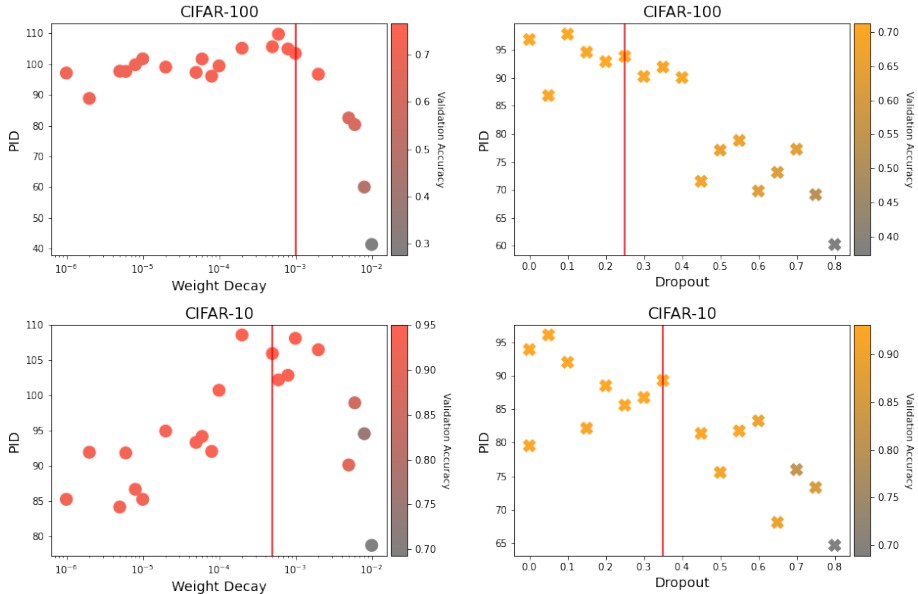

Figure 4: Relationship between PID and large values of regularization. Stronger red and yellow color indicates increased validation accuracy. Beyond a certain regularization strength, PID and validation accuracy decrease rapidly. The vertical red line in each plot indicates the regularization strength of the best performing model.

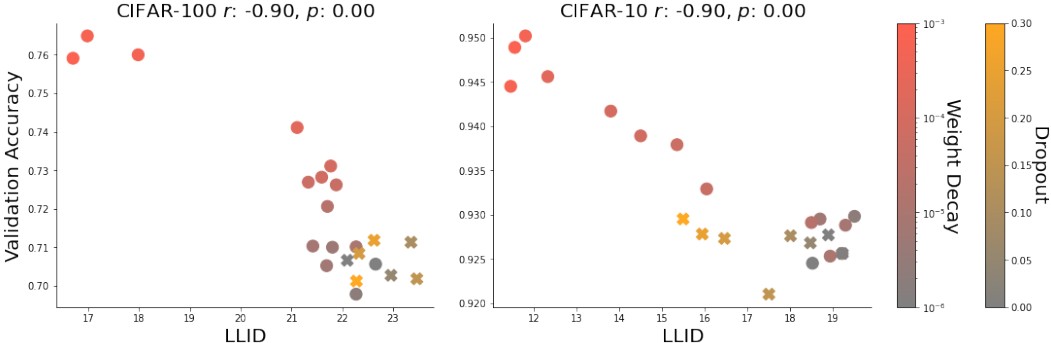

Figure 5: The correlation between LLID and validation accuracy for datasets CIFAR-100 (left) and CIFAR-10 (right) for various dropout and weight decay values.

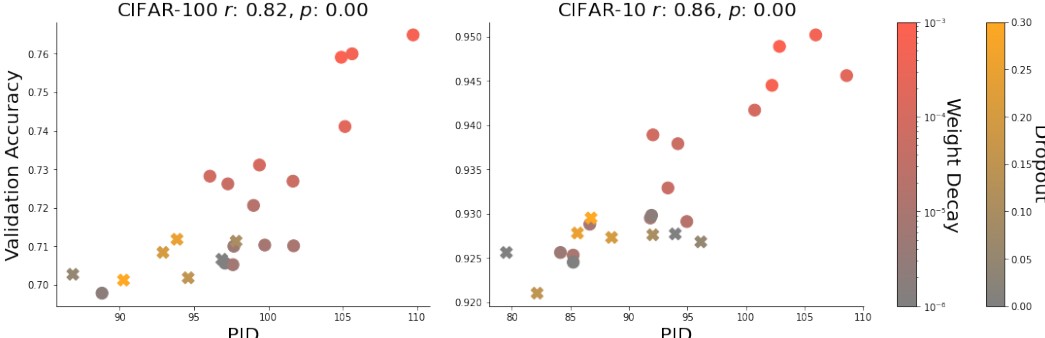

Figure 6: The correlation between PID and validation accuracy for datasets CIFAR-100 (left) and CIFAR-10 (right) for various dropout and weight decay values.

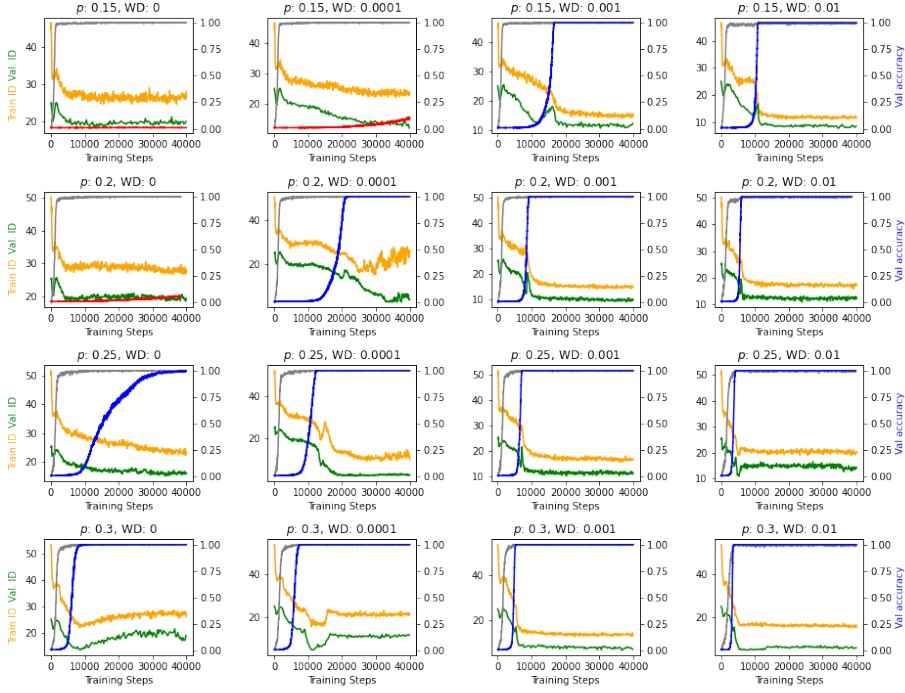

Figure 7: The values of train accuracy (shown in grey), validation accuracy (shown in blue and red), train LLID (shown in yellow) and LLID (shown in green) of transformer models throughout training on a modular division task. In addition to the trends observed in 3, train accuracy very quickly reaches 100% for all models and training LLID follows the same trends as validation LLID.

