# OpenReview forum: "Relating Regularization and Generalization through the Intrinsic Dimension of Activations"
_NeurIPS.cc/2022/Workshop/HITY — HITY Workshop NeurIPS 2022_

### Official Review · Reviewer_VXAA · 2022-10-05
**Empirical evidence that regularization decreases LLID and PID**

**Rating:** 0
**Confidence:** 4

**Review:**

The paper looks nice, but unfortunately I could not understand some central parts of the paper.

The paper's central claim is that the concept of "intrinsic dimension" (ID) can be used to capture how regularization and generalization relate -- either via last-layer ID (LLID) or through peak ID (PID). Unfortunately, these concepts are not well-defined and also couldn't find a definition in reference [3] which is cited. (The only thing I could understand from [3] is that the activations live in some kind of manifold, which has a dimension. Is this the ID?)

The authors define ID by pointing out that "it can be thought of as the minimal number of factors of variation in the model's representation of the data". This is however so cryptic, that I do not understand this. Accordingly I could also not understand what LLID, "the intrinsic dimension of the validation set's activations over all layers of the model", and PID, "the maximum intrinsic dimension of the validation set's activations over all layers of the model", are. I could not understand much of the rest of the paper because I did not understand these central definitions.

To make this paper readable, it would imo be necessary to define this concepts properly -- i.e. by a mathematical definition. The paper does not even try to develop a notation for the different layers of the neural network. I suggest to re-write the paper by introducing a mathematical notation and adding clear definitions.

---

### Official Review · Reviewer_Cnqu · 2022-10-11
**Very interesting paper**

**Rating:** 1
**Confidence:** 4

**Review:**

This paper uses the intrinsic dimension of layers (estimated by [1]) to show that the intrinsic dimension of the last layer correlates with the regularization applied. This metric can be used to measure a model's generalization capability. Further, they show that good generalization requires a trade-off between decreasing the intrinsic dimension of the last layer and keeping a high intrinsic dimension for the layer with the highest one.

The paper is clearly written and understandable throughout.
The topic fits well with this workshop, and the results might be of high interest to the whole community. Considering the introduced metric might pave the way to a better understanding of and new regularization methods.

What should be changed/added:
- There are no error bars/uncertainty intervals reported. Please repeat your experiments with multiple seeds to prove that your results are significant.
- Add a detailed description of the ID estimator you are using. One does not want to read an other full paper to understand what you are measuring.

I like that you added hyperlinks to almost all of your references! (Should become a new standard)

Typos, grammar, and formatting:
- l 13 to 16: hard to understand sentence
- Figure 3: Please add a description of the red curve to the figure. Took me a while to find it in the caption.
- l 348: Typo in 060

[1] E. Facco, M. d’Errico, A. Rodriguez, and A. Laio. Estimating the intrinsic dimension of
datasets by a minimal neighborhood information. Scientific Reports, 7(1):12140, Sep 2017.
ISSN 2045-2322. doi: 10.1038/s41598-017-11873-y. URL https://doi.org/10.1038/
s41598-017-11873-y.

---

### Official Review · Reviewer_A4Tr · 2022-10-14

**Rating:** 1
**Confidence:** 3

**Review:**

**Summary:** This paper empirically investigates the impact of regularization
techniques on the *intrinsic dimension* of model activations across the network
and how this quantity is connected to generalization.

**Strengths, Weaknesses & Questions:**
- The paper is well-written and nicely structured with a motivating
introduction. The experiments are well thought out and the observations are
explained and discussed comprehensibly.
- My main point of criticism: The intrinsic dimension is *the* central quantity
of your paper, but the definition you provide in lines 32-35 is quite vague and
therefore not sufficient in my opinion. I'd thus suggest adding a precise
mathematical definition and explanations of how you computed this quantity.
Without that, the reader is left with the information that it is some form of complexity
measure, which strongly limits in-depth interpretation and assessment of your
results.
- Figures 2, 5, and 6: Almost all weight decay values are shown in gray (I guess
because you use a log-style grid to generate them). This makes it hard to
distinguish different weight decay values. I'd thus suggest using the log scale
also on the color bar.
- Figure 4: I would have expected the relationship between PID and weight decay
to be monotonic. But, this does not seem to be the case: Very small weight decay
values *and* large values lead to small PID with a maximum PID in between. Do
you have an explanation for this behavior? Also, I'd suggest using a log scale
for the weight decay axes to spread out the small values.

**Minor:**
- Line 41: I don't understand what it means for activations to *live on* a
low-dimensional manifold.
- Figure 3: In my opinion, this Figure could be improved by using a 4x4 grid.
This way, the subplots could be ordered such that the $p$-value determines the
row and the WD-value the column. This would facilitate orientation and would
make it more obvious that the occurrence of red lines results from small $p$ and
small WD. Also, how is the y-label *Validation ID* defined - is it LLID?

---

### Decision · Program_Chairs · 2022-10-20

Accept